META-RESEARCH ARTICLE

# Assessment of transparency indicators across the biomedical literature: How open is open?

Stylianos Serghiou[1,2], Despina G. Contopoulos-Ioannidis[3], Kevin W. Boyack[4], Nico Riedel[5], Joshua D. Wallach[6], John P. A. Ioannidis[1,2,7,8,9]*

**1** Department of Epidemiology and Population Health, Stanford University School of Medicine, Stanford, California, United States of America, **2** Meta-Research Innovation Center at Stanford (METRICS), Stanford School of Medicine, Stanford, California, United States of America, **3** Division of Infectious Diseases, Department of Pediatrics, Stanford University School of Medicine, Stanford, California, United States of America, **4** SciTech Strategies, Inc., Albuquerque, New Mexico, United States of America, **5** Berlin Institute of Health, QUEST Center for Transforming Biomedical Research, Berlin, Germany, **6** Department of Environmental Health Sciences, Yale School of Public Health, New Haven, Connecticut, United States of America, **7** Stanford Prevention Research Center, Department of Medicine, Stanford University School of Medicine, Stanford, California, United States of America, **8** Department of Biomedical Data Science, Stanford University School of Medicine, Stanford, California, United States of America, **9** Department of Statistics, Stanford University School of Humanities and Sciences, Stanford, California, United States of America

* jioannid@stanford.edu

**Data Availability Statement:** Data are available on Open Science Framework at http://www.doi.org/10.17605/OSF.IO/E58WS.

## Abstract

Recent concerns about the reproducibility of science have led to several calls for more open and transparent research practices and for the monitoring of potential improvements over time. However, with tens of thousands of new biomedical articles published per week, manually mapping and monitoring changes in transparency is unrealistic. We present an open-source, automated approach to identify 5 indicators of transparency (data sharing, code sharing, conflicts of interest disclosures, funding disclosures, and protocol registration) and apply it across the entire open access biomedical literature of 2.75 million articles on PubMed Central (PMC). Our results indicate remarkable improvements in some (e.g., conflict of interest [COI] disclosures and funding disclosures), but not other (e.g., protocol registration and code sharing) areas of transparency over time, and map transparency across fields of science, countries, journals, and publishers. This work has enabled the creation of a large, integrated, and openly available database to expedite further efforts to monitor, understand, and promote transparency and reproducibility in science.

## Introduction

Research reproducibility [1] is a fundamental tenet of science, yet recent reports suggest that reproducibility of published findings should not be taken for granted [2]. Both theoretical expectations [3] and empirical evidence [4–8] suggest that most published results may be either non-reproducible or inflated. Cardinal among recommendations for more credible and efficient scientific investigation [9–18] is the need for transparent, or open science [16,19]. Such transparent practice, among others, facilitates reproducibility by providing the data and

**Funding:** This work was primarily funded by the National Institutes of Health award HHSN271201800033C to SciTech (K.B.) and METRICS (J.P.A.I). METRICS has also been supported by grants from the Laura and John Arnold Foundation. S.S. has been funded by the Department of Epidemiology and Population Health at Stanford University and as a Scholar of the Stanford Data Science Initiative. In the past 36 months, J.D.W. received research support through the Collaboration for Research Integrity and Transparency from the Laura and John Arnold Foundation and through the Center for Excellence in Regulatory Science and Innovation (CERSI) at Yale University and the Mayo Clinic (U01FD005938). The funders had no role in study design, data collection and analysis, decision to publish, or preparation of the manuscript.

**Competing interests:** The authors have declared that no competing interests exist.

**Abbreviations:** COI, Conflict of interest; NIH, National Institutes of Health; NLM, National Library of Medicine; NPV, negative predictive value; OCC, Open Citation Collection; OSF, Open Science Framework; PMC, PubMed Central; PMCID, PubMed Central ID; PMCOA, PubMed Central Open Access; PMID, PubMed ID; PPV, positive predictive value; STROBE, Strengthening the Reporting of Observational Studies in Epidemiology; TRIPOD, Transparent Reporting of a multivariable prediction model for Individual Prognosis Or Diagnosis; WOS, Web of Science.

code required to rerun reported analyses and promotes replicability by providing a detailed account of the methods and protocols employed.

In 2014, a team of experts outlined a number of recommended indicators of transparency that may be monitored by interested stakeholders to assess and promote improvements in biomedical research [20]. Our group previously evaluated 441 randomly selected biomedical articles from 2000 to 2014 [21], illustrating that across several of these indicators [21], only 136 (30.8%) articles had a conflict of interest (COI) disclosure (about the presence or absence of COI), and 213 (48.3%) had a funding disclosure (about presence or absence of funding); no article made its data or code openly available. A follow-up study of 149 articles published between 2015 and 2017 noted substantial improvements over time, with 97 (65.1%) sharing a COI disclosure, 103 (69.1%) sharing a funding disclosure, and 19 (12.8%) sharing at least some of their data; still, no code sharing was identified [22]. While our previous studies suggest low, but slowly improving levels of transparency and reproducibility, they are limited in that they only capture a small random subset of the biomedical literature and require laborious manual screening and data abstraction. A larger, high-throughput evaluation would be necessary to keep up to date with the pace of expanding literature and understand the distribution of transparency across more granular categories, such as time or fields of science.

Currently available tools can identify certain indicators of transparency, but they cannot be used to map and monitor these indicators across the published biomedical literature, their code is not openly available, their true performance is unknown, or they are paid services [23–26]. A recent publication utilized methods of machine learning to identify, among others, conflicts of interest and funding disclosures, but it can only process acknowledgments [27]. Another recently published tool, called SciScore, uses a machine learning method known as conditional random fields to identify measures of rigor (e.g., randomization, blinding, and power analysis) across the open access literature on PubMed and create a score of rigor and transparency. However, this tool does not include indicators of transparency (e.g., data or code sharing), did not provide article-specific data, and the underlying code was not made openly available [28].

Our work aims to expand our assessment of multiple indicators of transparency across the entire open biomedical literature. We hereby develop, validate, and utilize a new set of rule-based tools to assess Data sharing, Code sharing, COI disclosures, Funding disclosures, and open Protocol registration across the entire open access literature on PubMed. By doing so, we are able to map all of these indicators across publishers, journals, countries, and disciplines. Finally, we make our tools and data openly available, such that these may serve as a unifying foundation upon which future work will build and expand.

## Results

### Manual assessment of transparency and reproducibility across 499 PubMed articles (2015 to 2018)

We started with a manual assessment of a random sample of 499 English language articles from PubMed published in recent years (2015 to 2018) (Table 1). COI disclosures and Funding disclosures were assessed in all 499 articles, whereas all other indicators of transparency and reproducibility were only assessed in the more relevant subset of 349 articles with empirical data (henceforth referred to as "research articles"). This work expands on a previous in-depth assessment of a random sample of PubMed articles [22], as more recent literature is deemed likely to have the highest rates of transparency and yield additional data for training automated algorithms focusing on current practices.

**Table 1. Characteristics of 499 randomly selected English language articles from PubMed between 2015 and 2018.**

| | | Number (%) |
|---|---|---|
| **Total PubMed articles** | | **499** |
| **Year of publication** | **2015** | 121 (24%) |
| | **2016** | 121 (24%) |
| | **2017** | 143 (29%) |
| | **2018** | 114 (23%) |
| **On PMC** | | 197 (40%) |
| **Area** | **Preclinical research** | 331 (66%) |
| | **Clinical research** | 18 (4%) |
| **Field of science** | **Medicine** | 274 (55%) |
| | **Health services** | 69 (14%) |
| | **Brain research** | 46 (9%) |
| | **Biology** | 46 (9%) |
| | **Other (*n* = 5)** | 62 (12%) |
| **Characteristics** | | **Median (IQR)** |
| | **Authors per article** | 4 (2–7) |
| | **Affiliations per article** | 5 (2–8) |
| | **Citation count** | 2 (0–5) |
| | **Journal Impact Factor** | 2.8 (2.0–4.3) |
| | **Altmetric Attention Score** | 0.5 (0.0–2.7) |

Missing values: Field of science (2, 1%), Affiliations per article (25, 5%), Citation count (6, 1%), and Journal Impact Factor (76, 15%). Field of science was taken from SciTech, Citation count from Crossref (May 30, 2019), Journal Impact Factor of 2018 from WOS, and Altmetric Attention Score from Altmetric (May 30, 2019).

IQR, interquartile range; PMC, PubMed Central; WOS, Web of Science.

Out of all 499 articles, 341 (68%) had a COI disclosure, and 352 (71%) had a Funding disclosure. Of 349 research articles, 68 (20%) had a Data sharing statement, 5 (1%) had a Code sharing statement, 246 (71%) had a COI disclosure, 284 (81%) had a Funding disclosure, 22 (6%) had an openly registered Protocol, 175 (50%) made a statement of Novelty (e.g., "report for the first time"), and 33 (10%) included a Replication component in their research (e.g., validating previously published experiments, running a similar clinical trial in a different population, etc.) (Fig 1A). Most articles with transparency statements or disclosures claimed no COIs (89% [218/246] of research articles with COI disclosures), and a substantial portion disclosed exclusive support from public sources (36% [103/284] of research articles with Funding disclosures), availability of data upon request (25% [17/68] of research articles with Data sharing), and registration on ClinicalTrials.gov (50% [11/22] of research articles with Protocol registration) (S1 Fig). In 8 (12%) of 68 articles with a Data sharing statement claiming that all data were available in the text, no raw data were found.

Using information found in PubMed records of these articles alone (as opposed to screening the full text), would have missed most records with indicators of transparency (S1 Table). Abstracts are very limited in scope, and PubMed records are not tailored to systematically capture these indicators, perhaps with the exception of funding. Still, PubMed records gave information on funding for merely 35% of articles, while funding disclosures were in fact present in 71% of them.

By utilizing data from previous studies [20,22], we observed a marked increase in the proportion of publications reporting COI disclosures (Fig 1B, S2 Table). An increase is also seen in the availability of Funding disclosures and Data sharing statements, even though

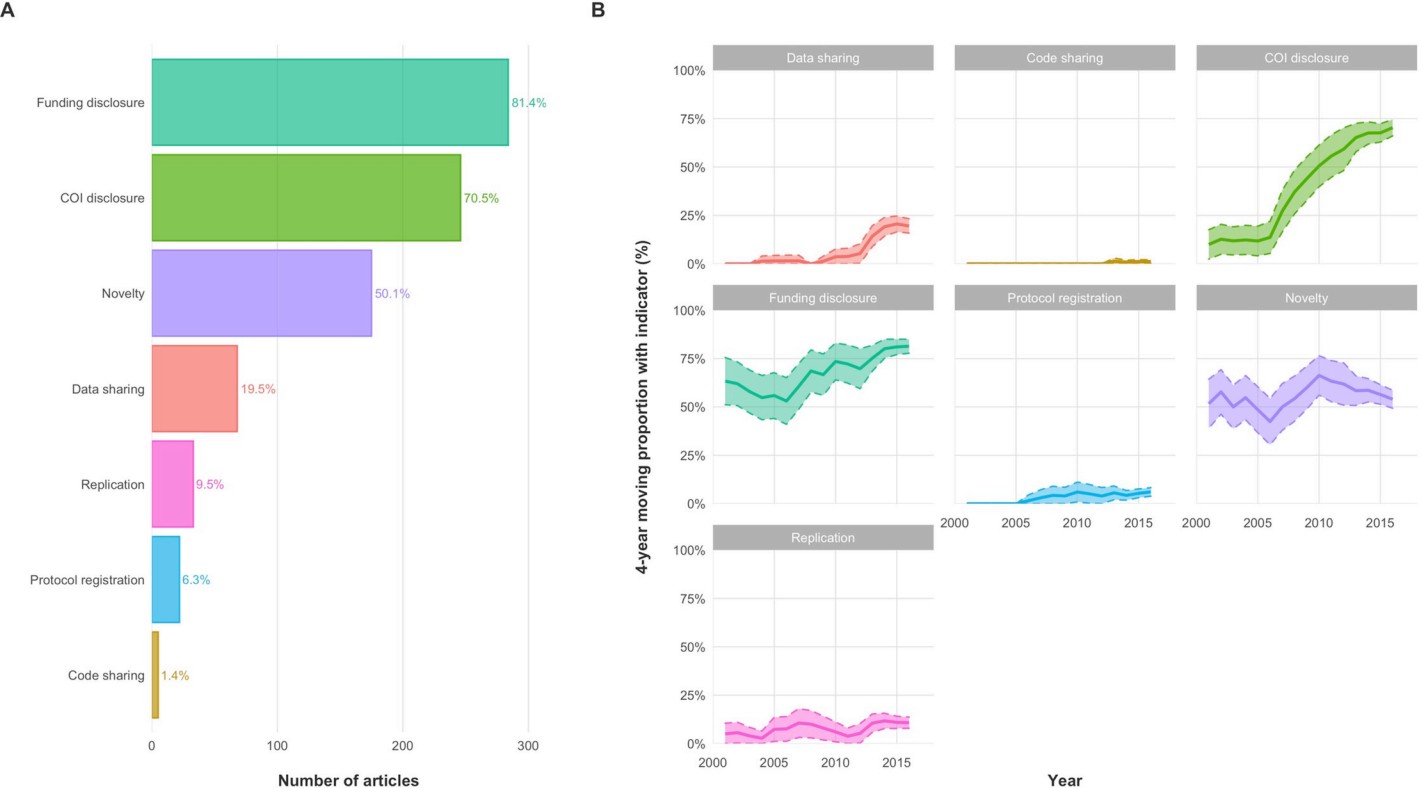

**Fig 1. Indicators of transparency and reproducibility across publications and time.** (A) Indicators of transparency and reproducibility across 349 research articles (2015–2018). Most publications included COI or Funding disclosures, but few mentioned Data, Code, or Protocol sharing. Similarly, most claimed Novelty but few mentioned a Replication component. (B) Indicators of transparency and reproducibility across time on the basis of manual assessment. These graphs merge data from this study on 349 research articles (2015–2018) with similar data from 2 previous studies on another 590 PubMed articles (2000–2018) [20,22]. Proportions are displayed as a 4-year centered moving average. The shaded region indicates the 95% CI. The most notable change is that of COI disclosures, the reporting of which increased from 12% in 2000 to 76% in 2018. The data underlying this figure can be found on OSF at http://www.doi.org/10.17605/OSF.IO/E58WS. CI, confidence interval; COI, Conflict of interest; OSF, Open Science Framework.

uncertainty is too large to claim trends. For Code sharing and Protocol registration, the numbers have remained low. No apparent trends were observed in statements of Novelty and Replication.

In our sample, reporting of COI disclosures, Funding disclosures, Data sharing, and Code sharing were significantly more frequently seen in articles available on PubMed Central (PMC) (S3 Table). Preclinical research articles reported Funding disclosures more often than clinical research articles (91% versus 70%), and clinical trials reported Protocol registration substantially more frequently than the rest (67% versus 3%).

## Automated assessment of transparency: Development and validation across 6,017 PMC articles (2015 to 2019)

We proceeded to test the performance of algorithms automating the process of identifying indicators of transparency by using a rule-based approach of regular expressions. Three were developed from scratch (COI disclosure, Funding disclosure, and Protocol registration), and two were adopted from an already existing library [29] (Data and Code sharing) and customized to enhance efficiency (see Materials and methods). Note that even though in the manual assessment Data sharing and Code sharing indicators capture any statement about data or code availability, in the automated assessment, we only capture newly generated open raw data

| | Data | Code | COI | Funding | Registration |
|---|---|---|---|---|---|
| Specificity | 98.6% (97.6%-99.5%) | 99.7% (99.6%-99.9%) | 99.5% (98.5%-100.0%) | 98.1% (96.2%-99.5%) | 99.7% (99.5%-99.8%) |
| Sensitivity | 75.8% (61.4%-93.9%) | 58.7% (34.0%-93.7%) | 99.2% (98.6%-99.8%) | 99.7% (99.3%-99.9%) | 95.5% (91.9%-98.6%) |
| PPV | 93.0% (88.0%-97.0%) | 88.2% (81.7%-93.8%) | 99.9% (99.7%-100.0%) | 99.7% (99.3%-99.9%) | 92.1% (88.3%-95.9%) |
| NPV | 94.4% (89.1%-98.9%) | 98.6% (96.2%-99.9%) | 96.8% (94.4%-99.1%) | 98.1% (96.2%-99.5%) | 99.8% (99.7%-99.9%) |
| Accuracy | 94.2% (89.7%-97.9%) | 98.3% (96.0%-99.6%) | 99.3% (98.8%-99.7%) | 99.4% (99.0%-99.8%) | 99.5% (99.3%-99.7%) |
| Prevalence (true) | 19.5% (15.6%-24.4%) | 3.4% (2.0%-5.7%) | 81.0% (79.9%-82.1%) | 84.6% (83.5%-85.5%) | 4.0% (3.5%-4.5%) |
| Prevalence (estimated) | 15.9% (14.4%-17.7%) | 2.3% (1.8%-2.8%) | 80.5% (79.4%-81.5%) | 84.6% (83.5%-85.5%) | 4.2% (3.6%-4.7%) |
| Error | 3.6% (-0.0%-8.1%) | 1.1% (-0.2%-3.4%) | 0.5% (0.1%-1.1%) | 0.0% (-0.4%-0.4%) | -0.1% (-0.4%-0.1%) |

**Fig 2. Validation of algorithms for Data sharing, Code sharing, COI disclosure, Funding disclosure, and Protocol registration in 6,017 PMC articles from 2015 to 2019.** The displayed performance was assessed in subsamples from 6,017 PMC articles: 189 research articles for Data sharing (100 positive, 89 negative), 291 research articles for Code sharing (110 positive, 181 negative), 325 articles for COI disclosure (100 positive, 225 negative), 326 for Funding disclosure (100 positive, 226 negative), and 308 for Protocol availability (161 positive, 147 negative). All algorithms displayed high accuracy (>94%) and low error in prevalence estimation (≤3.6%) compared to manual assessment. Error, difference between true and estimated prevalence; NPV, negative predictive value; PPV, positive predictive value (precision); Prevalence (true), manual estimate of proportion of articles with indicator; Prevalence (estimated), automated estimate of proportion of articles with indicator. The data underlying this figure can be found on OSF at http://www.doi.org/10.17605/OSF.IO/E58WS. COI, Conflict of interest; OSF, Open Science Framework; PMC, PubMed Central.

or code to align with previous work in this field [29]. In a random unseen sample of 6,017 PMC records from 2015 to 2019, these algorithms predicted Data sharing in 764 (13%) records, Code sharing in 117 (2%), COI disclosure in 4,792 (80%), Funding disclosure in 5,022 (84%), and Protocol registration in 261 (4%); 61 (1%) claimed both Data and Code sharing. Examples for the presence of any of those indicators can be found in the Supporting information (S2 Fig).

In the tested samples of 6,017 PMC articles, the accuracy, positive predictive value (PPV), and negative predictive value (NPV) of all 5 algorithms was ≥88% (Fig 2). Even though all algorithms were highly specific (>98%), the data and code sharing algorithms were not as sensitive as the rest (76% and 59%, respectively, versus >95% for all other indicators). This was in part because algorithms did not identify raw data made available as supplements or did not

recognize less popular code repositories (Bitbucket and Open Science Framework [OSF]). The estimated sensitivity of the code sharing algorithm was particularly driven by a single study that provided its code as a supplement, but was falsely labeled negative (see Code Sharing in S2 Text). As such, the algorithm made 1 mistake in 88 manually assessed research articles with no Data or Code sharing. However, with the vast majority of articles not sharing data or code (5,197/6,017), this 1 article was dramatically overweighted. This is reflected by the large confidence interval (34% to 94%), which includes the estimated sensitivity of 73% from a random sample of 800 PMC research articles of 2018 calculated by the original authors of this algorithm [29].

The difference between manually adjudicated indicator prevalence (true) versus machine-adjudicated prevalence (estimated) was ≤1.1% for all algorithms other than for Data sharing, for which it was 3.6%. By design, the COI disclosure, Funding disclosure, and Protocol registration algorithms had 100% sensitivity and specificity in the training set of 499 articles (2015 to 2018); the Data and Code sharing algorithms had 76% sensitivity and 98% specificity in a training set of 868 random PubMed articles from 2015 to 2017 [29]. Detailed assessment of each algorithm can be found in the Supporting information (S2 Text).

## Automated assessment of transparency: Transparency across 2.75 million PMCOA articles (1959 to 2020)

We then proceeded to test the entire PubMed Central Open Access (PMCOA) subset using these algorithms. We identified 2,751,484 records as of February 29, 2020, of which 2,751,420 were unique (Table 2). For each record, we extracted 158 variables, of which 39 were metadata (35 from PMC, 2 from National Institutes of Health (NIH) Open Citation Collection (OCC), 2 from Web of Science (WOS), and 1 from SciTech) (S4 Table), and 119 were related to the indicators of transparency (5 for presence/absence of each indicator, 5 with extracted text for each indicator, and 93 indicating which aspects of text were found relevant for each indicator) (S5 Table). Of these, 2,285,193 (83%) had been labeled as research articles by OCC, and 2,498,496 (91%) were published from 2000 onwards in a total of 10,570 different journals. Note that open access records from PMC did not include any supporting information and that the classification of OCC into research articles is imperfect (see "Sources of metadata" in Materials and methods).

Out of 2,751,420 open access PMC records (1959 to 2020), our algorithms identified mentions of Data sharing in 243,783 (8.9%), Code sharing in 33,405 (1.2%), COI disclosure in 1,886,907 (68.6%), Funding disclosure in 1,858,022 (67.5%), and Protocol registration in 70,469 (2.6%) (Fig 3A, S6 Table). Adjusting for the discrepancies between predicted and true estimates seen in the validation set, we estimate that Data sharing information was mentioned in 14.5% (95% CI, 11.0% to 18.8%), Code sharing information in 2.5% (95% CI, 1.2% to 4.7%), COI disclosure information in 69.5% (95% CI, 69.0% to 70.1%), Funding disclosure information in 67.9% (95% CI, 67.6% to 68.3%), and Protocol registration information in 2.5% (95% CI, 2.5% to 2.6%). The majority of COI disclosures reported no conflicts of interest (1,531,018; 81%), and the majority of Funding disclosures reported receipt of funds (1,675,513; 90%). Associations between indicators and literature characteristics are reported in the Supporting information (S3 Text, S7 Table). Note that unlike in our manual assessment, the numbers reported in this section refer to the entire literature on PMCOA, not merely research articles; all analyses were repeated in research articles alone with no meaningful changes.

## Transparency across time, countries, fields of science, journals, and publishers

Considering the 2,751,420 open access records on PMC (PMCOA) (1959 to 2020), over time, all indicators have experienced an upward trend in reporting (Fig 3B). However, this increase

**Table 2.** Metadata across all 2,751,420 open access publications on PubMed Central (PMCOA).

| | | Number (%) |
|---|---|---|
| **Open access PMC articles** | | **2,751,420** |
| **Research** | | 2,285,193 (83%) |
| **Country** | USA | 126,256 (26%) |
| | China | 80,667 (17%) |
| | UK | 68,413 (14%) |
| | Other (*n* = 2,113) | 312,302 (65%) |
| **Field of science** | Medicine | 1,175,753 (50%) |
| | Health services | 299,299 (13%) |
| | Biology | 261,326 (11%) |
| | Other (*n* = 9) | 1,015,042 (37%) |
| **Characteristics** | | **Median (IQR)** |
| | Year of publication | 2015 (2012–2018) |
| | Authors per article | 5 (3–7) |
| | Figures per article | 3 (0–5) |
| | Tables per article | 1 (0–3) |
| | References per article | 32 (15–50) |
| | Citation count | 6 (2–15) |
| | Journal Impact Factor | 3.0 (2.4–4.4) |

Note that Country contains more than the total number of countries because many publications report locations other than country (e.g., state, city, etc.). Similarly, even though we have gone at great length to standardize names of journals (see Materials and methods), the reported number is likely an overestimate. Missing values: Research (240,244, 9%), Country (2,271,577, 83%), Field of science (396,331, 14%), Year (21,080, 1%), Citation count (870,186; 32%), and Journal Impact Factor (909,098, 33%). Note that the large number of missing data for Citation count and Field of science are due to many articles not having a PMID (some articles are on PMC but are not admitted to PubMed). Field of science was taken from SciTech, Citation count from OCC (April 17, 2020), and Journal Impact Factors of 2018 from WOS.

IQR, interquartile range; OCC, Open Citation Collection; PMC, PubMed Central; PMCOA, PubMed Central Open Access; PMID, PubMed ID; WOS, Web of Science.

has been far more dramatic for COI and Funding disclosures than for Data sharing, Code sharing, and Protocol registration. Specifically, all indicators have risen from approximately 0% in 1990 to an estimated 15% in 2020 for Data sharing, 3.1% for Code sharing, 90% for COI disclosures, 85% for Funding disclosures, and 5% for Protocol registration. For research articles, the respective proportions for 2020 are 17%, 3.5%, 91%, 89%, and 5.7%, respectively. The proportion of publications reporting on these indicators was homogeneous across countries, with most reporting COI and Funding disclosures, but only the minority reporting Data sharing, Code sharing, or Protocol registration (Fig 4).

In terms of field of science (see S8 Table for examples), all fields reported COI and Funding disclosures more frequently than other indicators of transparency (Fig 5A; see S9 Table for example phrases across fields and indicators). However, publications from different fields tended to report indicators of transparency at substantially different proportions. For example, publications classified within Biology or Infectious diseases tended to share Data (29% and 17%, respectively) and Funding disclosures (90% and 86%, respectively) more than publications within Medicine (7% for data, 71% for funding) or Health services (3.6% for data, 68% for funding). On the contrary, publications within Health services were more likely than other fields of science to share COI disclosures (81.4% versus 77.4%; 95% CI of difference, 3.9% to

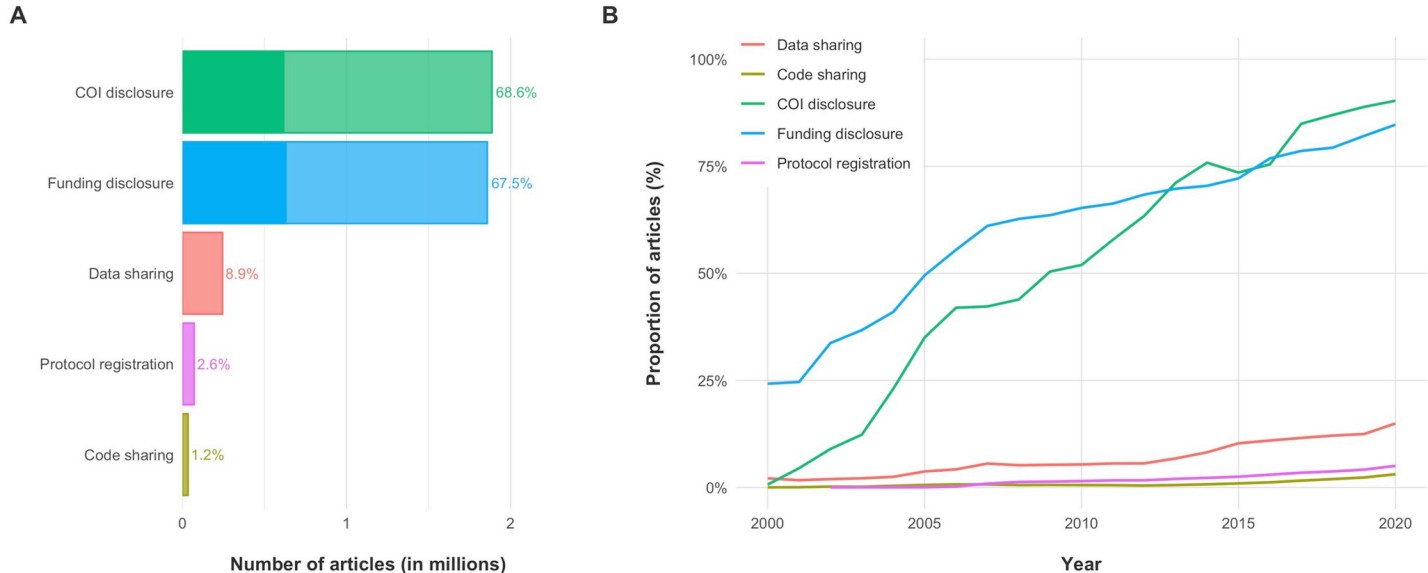

**Fig 3. Indicators of transparency across the entire open biomedical literature on PMC (PMCOA) and time.** (A) Most open biomedical articles report COI disclosures and Funding disclosures, but only a minority report any Data sharing, Code sharing, or open Protocol registration. Note that this figure displays the obtained results with no adjustments (see text for adjustments). The opaque section of the bars for COI and Funding disclosure denotes the number of publications that would have been recognized had we only used information currently provided on PMC. Such information appears to underestimate the true prevalence of these indicators by two-thirds. (B) Transparency in the open biomedical literature on PMC (2000–2020). Reporting on all indicators of transparency has been increasing for the past 20 years. However, the increase for COI and Funding disclosures has been much more dramatic than for Data sharing, Code sharing, and Protocol registration. The data underlying this figure can be found on OSF at http://www.doi.org/10.17605/OSF.IO/E58WS. COI, Conflict of interest; OSF, Open Science Framework; PMC, PubMed Central; PMCOA, PubMed Central Open Access.

4.2%) or practice open Protocol registration (5.7% versus 2.5%; 95% CI of difference, 3.1% to 3.3%).

We then proceeded to further map indicators of transparency across galaxies of science (Fig 5B). These galaxies were previously developed using 18.2 million PubMed articles published between 1996 and 2019 and divided into approximately 28,000 clusters of articles [30]. Each cluster comprises similar articles and is colored according to the field associated with the most prevalent journals within the cluster. The size of each colored cluster was modified to reflect the proportion of articles published between 2015 and 2019 that are open access (1,490,270 articles) or report Data sharing (177,175 articles), Code sharing (25,650 articles), COI disclosure (1,263,147 articles), Funding disclosure (1,192,4996 articles), or Protocol registration (52,798 articles). The most recent 5 years cover more than half of the articles on PMCOA and were chosen to most accurately portray the current practice of transparency. These galaxies further corroborate that most recent open access articles share COI and Funding disclosures, but do not share Data, Code, or Protocol.

Out of 2,477 journals with at least 100 articles on PMCOA between 1990 and 2020, the majority consistently reported COI disclosures (42% of journals reported them in ≥90% of their publications; 64% in ≥70% of their publications) and Funding disclosures (21% of journals reported them in ≥90% of their publications; 54% in ≥70% of their publications), but only the minority reported consistently on Data sharing (0.1% of journals in ≥90% of their publications; 0.5% of journals in ≥70% of their publications), Protocol registration (only 1 journal in ≥70% of its publications), or Code sharing (no journal in ≥70% of its publications; highest percentage was 40% in GigaScience) (Fig 6A). However, 79% of these journals have shared data, 71% have shared a protocol, and 39% have shared code at least once (i.e., 21%,

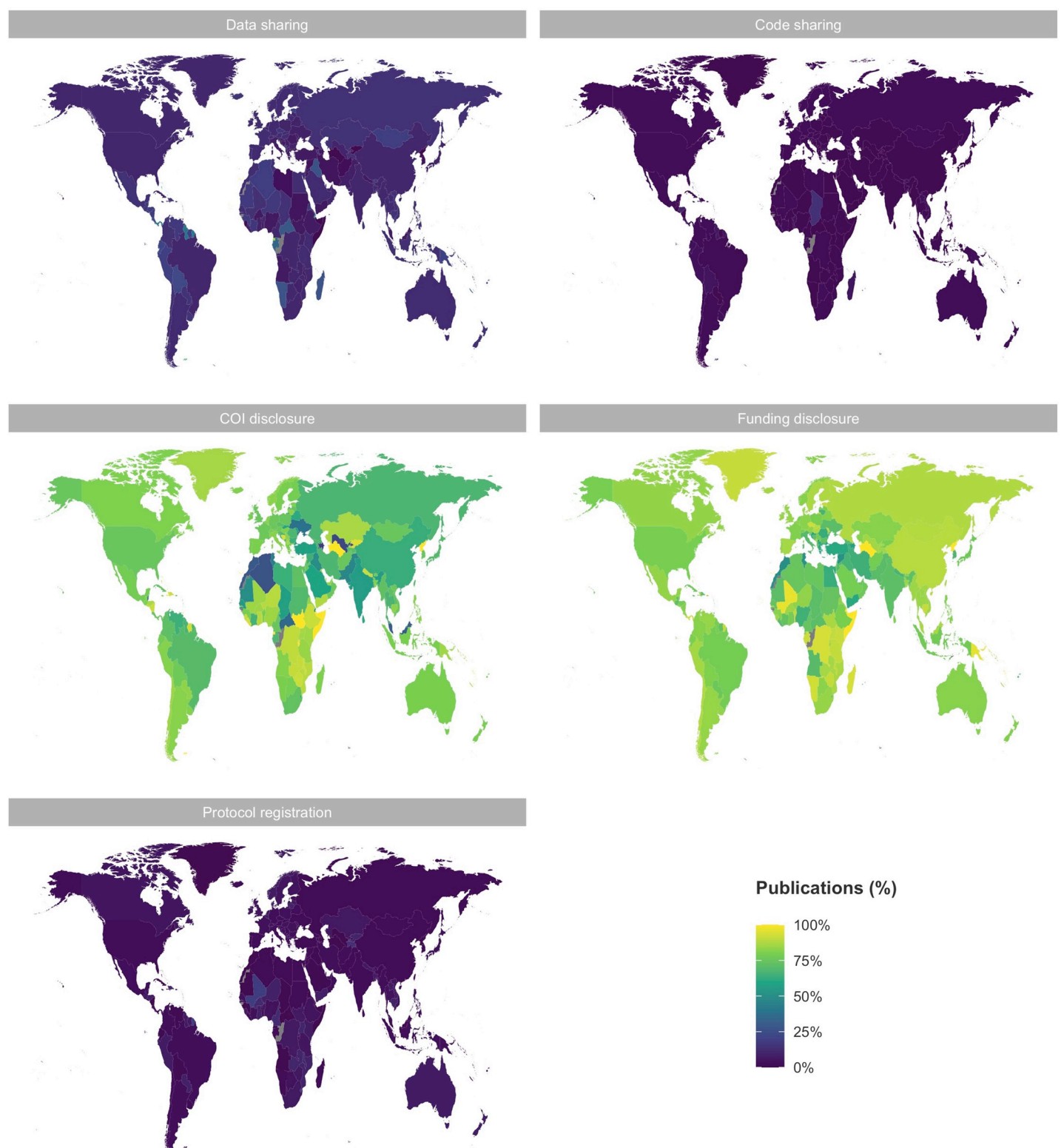

**Fig 4. Indicators of transparency across countries of affiliation for all open access articles on PMC (PMCOA) for 1990–2020.** Indicators of transparency are roughly homogeneously reported across countries. The light green to yellow maps indicate that the majority of publications from most countries reported COI and Funding disclosures. The purple maps indicate that the majority of publications from most countries did not report any Data, Code, or Protocol sharing. Note that the country was only reported in 479,843 articles. The data underlying this figure can be found on OSF at http://www.doi.org/10.17605/OSF.IO/E58WS. COI, Conflict of interest; OSF, Open Science Framework; PMC, PubMed Central; PMCOA, PubMed Central Open Access.

**A**

| | Data sharing | Code sharing | COI disclosure | Funding disclosure | Protocol registration | Total |
|---|---|---|---|---|---|---|
| MEDICINE | 81338 (7.0%) | 10990 (0.9%) | 924502 (79.4%) | 829086 (71.2%) | 41029 (3.5%) | 1,886,945 |
| HEALTH | 10587 (3.6%) | 1456 (0.5%) | 240968 (81.3%) | 201450 (68.0%) | 17062 (5.8%) | 471,523 |
| BIOLOGY | 76152 (29.3%) | 10125 (3.9%) | 183235 (70.5%) | 233847 (90.0%) | 275 (0.1%) | 503,634 |
| INF DIS | 39447 (17.0%) | 3479 (1.5%) | 171330 (73.8%) | 199996 (86.2%) | 2501 (1.1%) | 416,753 |
| BRAIN | 11882 (6.2%) | 2835 (1.5%) | 154760 (80.9%) | 148857 (77.9%) | 6285 (3.3%) | 324,619 |
| CHEMISTRY | 9778 (7.1%) | 831 (0.6%) | 99052 (72.2%) | 117892 (85.9%) | 528 (0.4%) | 228,081 |
| ENGNG | 1095 (4.8%) | 262 (1.1%) | 18196 (79.6%) | 19658 (86.0%) | 22 (0.1%) | 39,233 |
| SOC SCI | 1291 (7.9%) | 348 (2.1%) | 12495 (76.3%) | 12287 (75.0%) | 541 (3.3%) | 26,962 |
| COMP SCI | 743 (6.7%) | 518 (4.7%) | 8942 (80.3%) | 9379 (84.2%) | 48 (0.4%) | 19,630 |
| PHYS/MATH | 181 (7.1%) | 52 (2.0%) | 1503 (59.1%) | 2184 (85.9%) | 0 (0.0%) | 3,920 |
| HUMANITIES | 8 (2.1%) | 2 (0.5%) | 153 (40.1%) | 206 (53.9%) | 0 (0.0%) | 369 |
| EARTH | 7 (4.8%) | 2 (1.4%) | 41 (28.3%) | 141 (97.2%) | 0 (0.0%) | 191 |

**B**

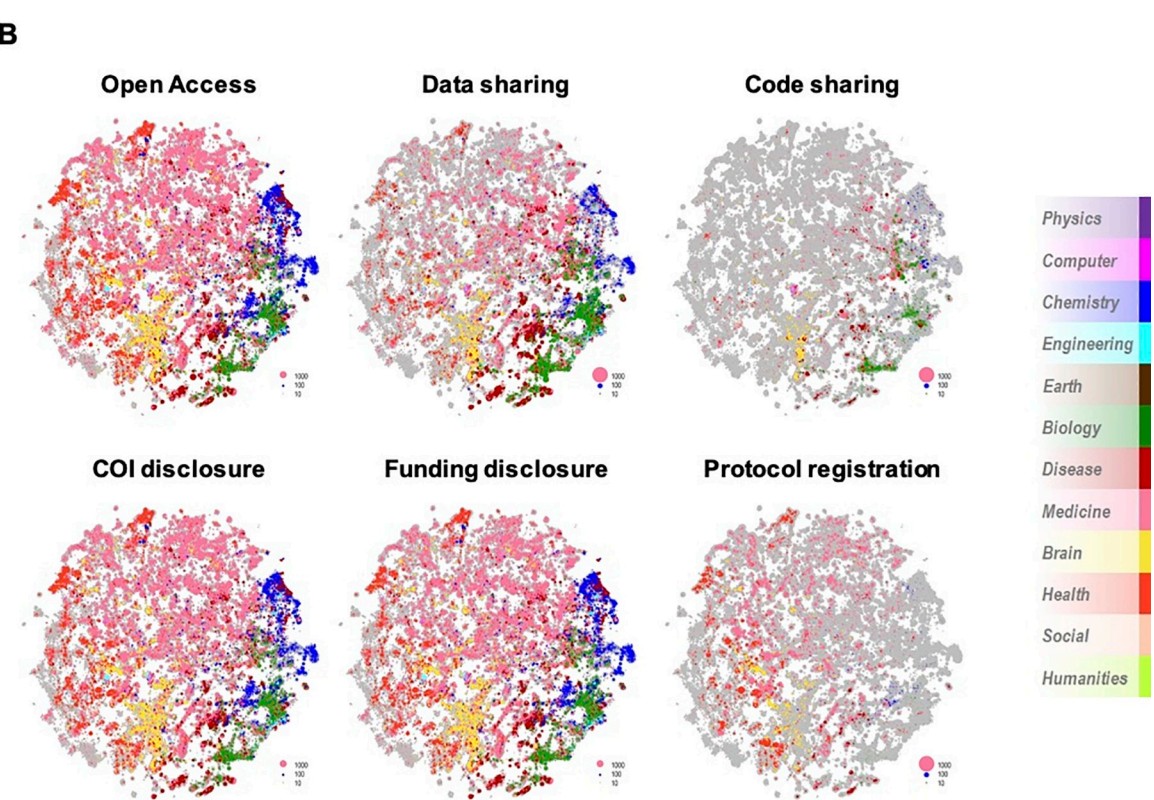

**Fig 5. Indicators of transparency across fields of science on PMCOA.** (A) Reporting of indicators of transparency across fields of science for all articles on PMCOA since 1990. COI and Funding disclosures are by far the most highly reported indicators within all fields, whereas Code sharing and Protocol registration are by far the least reported. The indicator with the biggest proportional difference between minimum and maximum reporting across fields of science was Protocol registration (0.1% versus 5.8%; Coefficient of variation, 130), and the indicator with the smallest proportional difference was Funding disclosure (53.9% versus 90.0%; Coefficient of variation, 14.5). (B) Indicators of transparency in all articles of PMCOA published between 2015 and 2019 across galaxies of science. The galaxy in gray represents all clusters of articles published between 2015 and 2019. On top of the gray galaxy, we overlaid colored representations of the proportion of each cluster that is open access or reports on any of the indicators of transparency. The open access galaxy is very similar to that of COI and Funding disclosures, suggesting that most of the open literature reports on both. A number of Chemistry (blue) and Biology (green) clusters are smaller in COI disclosure, whereas a number of Health services (red) and Infectious diseases (burgundy) clusters are smaller in Funding disclosure. Biology (green) and Infectious diseases (burgundy) are pronounced in Data sharing. A very small proportion of clusters report Code sharing or open Protocol registration—of those, the majority are Biology (green) and Health services (red) clusters, respectively. The data underlying this figure can be found on OSF at http://www.doi.org/10.17605/OSF.IO/E58WS. COI, Conflict of interest; OSF, Open Science Framework; PMCOA, PubMed Central Open Access.

29%, and 61% have never shared any data, protocol, or code, respectively). These numbers did not change meaningfully when considering research articles alone.

Out of 609 publishers with at least 100 articles (research or non-research) on PMCOA between 1990 and 2020, the majority consistently reported COI disclosures (30% had disclosures in at least 90% of articles; 49% had disclosures in at least 70% of articles) and Funding disclosures (10% had disclosures in ≥90% of articles; 28% had disclosures in ≥70%), but only the minority reported consistently on Data sharing (1 publisher in ≥70%), Protocol registration (0 in ≥70%), or Code sharing (0 in ≥70%) (Fig 6B). However, 71.4% of publishers have shared a protocol, 70.9% have shared data, and 30% have shared code at least once (i.e., 28.6%, 29.1%, and 70.0% of publishers have never shared protocol, data, or code, respectively). These numbers did not change meaningfully when considering research articles alone.

## Discussion

Our evaluation of 2,751,420 open access PMC articles (1959 to 2020) suggests that there have been substantial improvements in reporting of COI and Funding disclosures across time, fields of science, and publication venues, but that Data sharing, Code sharing, and Protocol registration are significantly lagging. It also illustrates that using automated approaches to study and understand the biomedical literature is possible and can yield insights over and above those possible with manual assessments of small random samples. This effort has led to the creation of a database of all open access biomedical literature (2.75 million articles), alongside granular information about indicators of transparency and metadata integrated from PMC, OCC, and SciTech that is now openly available (see our Data Availability Statement). We envision that this resource will encourage future efforts trying to further improve on our openly available tools and understand the uptake of transparency and reproducibility practices and their impact on science.

The currently presented algorithms were able to extract relevant text and by construction make multiple pieces of information about their predictions available. For example, in the case of COI disclosures, the algorithm may label a text as referring to no conflicts of interest present, commercial interests present, financial disclosures present, receipt of consulting fees, etc. As such, they can facilitate other teams in building systems on top of this with more nuanced definitions of COI without having to retrain and recalibrate complex black box systems.

Indeed, we hope that our tools, data, and results set a precedent that can unify, inspire, and inform future research and practice. Our tools have been open sourced as an R package so that other teams or individuals may apply them within their collection of articles, improve on their current implementation, and expand them to indicators of transparency not currently captured; all within 1 unified package. Our data, as indicated above, provide a foundational yet

**A**

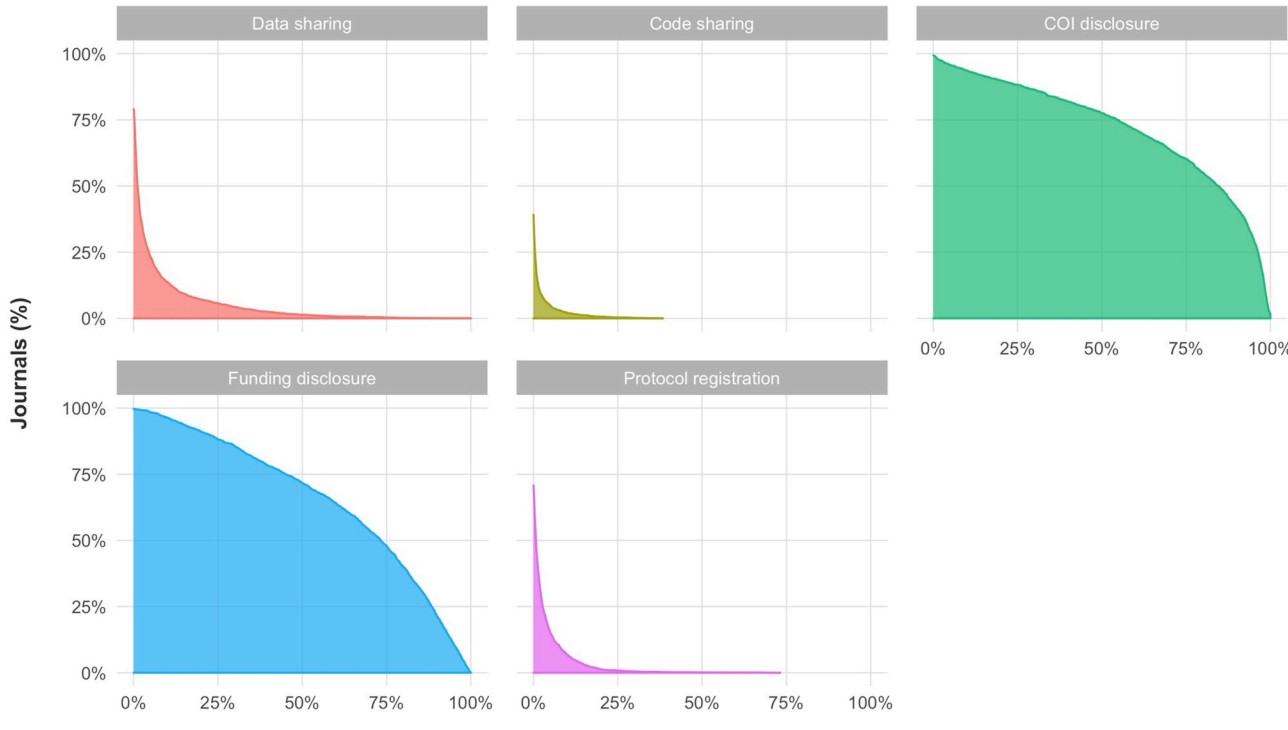

**B**

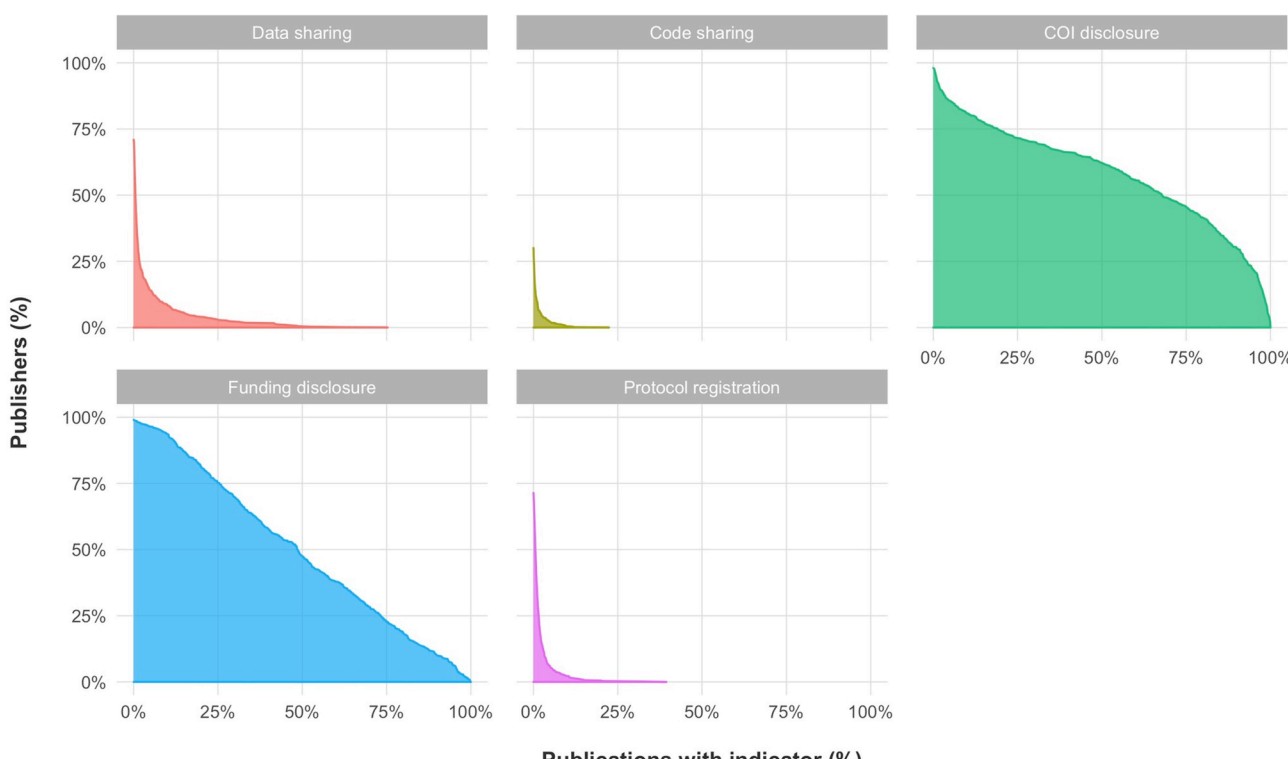

**Fig 6. Indicators of transparency across 2,477 journals and 609 publishers with at least 100 publications on PMCOA between 1990 and 2020.** (A) Proportion of journals with at least a designated proportion of publications reporting on each indicator of transparency. For example, for the graph on Funding disclosures, approximately 50% of journals (vertical axis) report Funding disclosures in at least approximately 75% of their publications (horizontal axis). Similarly, they indicate that no journal reports Protocol registration in more than approximately 75% of publications. The concave (bent outward) distribution of COI and Funding disclosures indicates that most publications in most journals disclose COI or Funding, respectively. However, the convex (bent inward) distribution of Data sharing, Code sharing, and Protocol registration indicate that most publications do not report on each of these indicators in most journals. About three quarters of journals have hosted at least 1 publication sharing data, and about one-third of journals have hosted at least 1 publication sharing code. (B) Proportion of publishers with at least a designated proportion of articles (research or non-research) reporting on each indicator of transparency. For example, these graphs indicate that approximately 50% of journals (vertical axis) report Funding disclosures in at least approximately 50% of their articles (horizontal axis). Similarly, no publisher reports Data sharing in more than approximately 75% of articles. As in (A), the shapes of these distributions indicate that most articles of most publishers report on COI and Funding, but most do not report on Data sharing, Code sharing, or Protocol registration. Almost three quarters of publishers host articles that never report Code sharing, and roughly one-quarter of publishers host articles that never report Data sharing. The data underlying this figure can be found on OSF at http://www.doi.org/10.17605/OSF.IO/E58WS. COI, Conflict of interest; OSF, Open Science Framework; PMCOA, PubMed Central Open Access.

flexible map of transparency to expedite further research in the area, such that future researchers may investigate these indicators across different definitions or different subsets of the literature or integrate these data within their own data pipeline such that any re-extraction is avoided. Finally, our results help us monitor the status quo and suggest the existence of different practices across publishers and disciplines, providing an opportunity to reflect and learn from the experience of each other.

In that vein, our results provide an opportunity for publishers, journals, and the National Library of Medicine (NLM) to standardize reporting of such indicators. Even though the NLM provides XML tags for labeling aspects of an article in a machine-readable way [31], our data illustrate that when these exist (e.g., COI and Funding disclosures), they are used by a minority of journals and that these do not exist for many other important aspects of transparency (e.g., Protocol registration and Code sharing). Given our data, most journals use different language when referring to such indicators, and only a minority of journals seem to use appropriate tags to label transparency statements, and no journal used purpose-built tags for Protocol registration or Code sharing. A number of initiatives aim to convert PDFs to tagged files [32], but the institution of a journal-wide requirement for standard language and tagging would enormously reinforce these efforts. In the meantime, both PubMed and other journals can, if they so desire, use our openly available algorithms to improve the availability of information about transparency.

## Limitations

First, the performance of our algorithms was assessed in a recent sample of PMC. As such, we cannot be certain of their performance across a sample of articles outside PMC, older articles, or articles published in the future. Similarly, we cannot be certain that average performance is representative of performance within specific fields of science, journals, or publishers. However, the majority of articles in the published literature have been published after 2000, and the findings of our algorithms corroborate those seen in the current and previous manual assessments [20,22]. Second, we are only using the Open Access Subset of PMC. There are other freely available articles within as well as outside PMC that we did not assess. However, previous publications have illustrated that the Open Access Subset is of similar composition as the non-Open Access Subset of PMC [22], and PMC represents about half of the recently published literature available on PubMed. Third, algorithms in general do not guarantee that the statements identified are either accurate, full, or true. This is why we have designed our algorithms to not only extract text, but also indicate why an article was labeled as including information on an indicator. We hope that future work will study these questions further. Fourth, some of the XML versions used in our biomedical open literature–wide search do not include the COI

or Funding disclosures that were included in the published PDF. Nevertheless, given that the numbers identified are very similar to those of manual labeling, we do not believe that this is a significant concern. Finally, even though we have assessed all open access articles on PMC, some of these articles cannot be expected to share Data, Code, or a Protocol registration (e.g., an opinion paper). Even though future work is necessary to identify such contextual characteristics as the type of article, we have mitigated their impact by incorporating their existence within the evaluation of our algorithms and, where meaningful, providing assessments of indicators within research articles alone. However, it should be noted that many of the articles labeled by OCC as non-research may in fact contain empirical research (see "Sources of metadata" in Materials and methods).

Allowing for these caveats, we provide tools that can expedite the massive assessment of the scientific literature for the most important indicators of transparency and map such indicators across time, countries, journals, publishers, and disciplines. This may help reinforce efforts for making such transparency features even more routine across published articles and understand where and how critical resources may be invested in improving transparency in the biomedical literature.

## Materials and methods

This manuscript was prepared using guidance from the Strengthening the Reporting of Observational Studies in Epidemiology (STROBE) reporting guidelines [33] for observational studies and the Transparent Reporting of a multivariable prediction model for Individual Prognosis Or Diagnosis (TRIPOD) guidelines for reporting prediction models [34].

### Data sources

**PubMed.** First, we randomly assembled a retrospective cohort of 520 records made available on PubMed between 2015 and 2018. PubMed is maintained by the United States NLM and provides citations to 30,732,929 references of the biomedical and life sciences literature (as of March 2020), 4,552,825 of which were published between 2015 and 2018 [35]. All English articles published between 2015 and 2018 were eligible for analysis.

**PubMed Central.** PMC is a free full-text archive for a subset of the publications available on PubMed. It was set up in 2000 and includes publications from journals that have agreed to either share all of their publications, NIH-funded publications, or a select subset of their publications [35]. Out of 5,747,776 publications made available on PubMed between 2015 and 2019 (in terms of Entrez Date), 2,497,046 (43.4%) were also made available on PMC. We randomly identified and downloaded the PDF of 6,017 of these records.

**PubMed Central Open Access Subset.** As of February 29, 2020, out of 6,016,911 records ever made available on PMC (1795 to 2020), 2,754,689 are part of the PMCOA [36]; not all articles on PMC are part of PMCOA. These articles are made available under a Creative Commons or similar license, and their full text (but not their supplements) may be downloaded in bulk (as XML files) and used for research; these are the publications that were used to estimate the open access–wide biomedical literature degree of transparency.

**Sources of metadata.** We extracted all metadata related to the articles of interest provided by PubMed and PMC (e.g., journal of publication, publisher, authors, affiliations, etc.). For all manually assessed articles, we extracted all social media–related data from Altmetric on May 30, 2019. Altmetric captures and tracks the use and sharing of articles across social media. For all manually assessed articles, we extracted citation counts from Crossref on May 30, 2019; for automatically assessed articles, we extracted citation counts from the NIH OCC (iCite 2.0) [37] on April 17, 2020. OCC is a recent initiative of the NLM, which attempts to map all

citations from PubMed records to PubMed records and make them openly available. Note that OCC only has citation data for articles with a PubMed ID (PMID), so for articles with a PubMed Central ID (PMCID) but no PMID, we had no citation data. We also used OCC to extract whether an article is considered a research article or not—this is based on definitions by PubMed and considers articles such as those labeled by PubMed as journal articles, randomized controlled trials, or observational studies as research articles and articles such as those labeled as reviews, editorials, news, or comments as non-research [38]—it should be noted that in our work we identified that PubMed annotations do not always agree with PMC and that 47,631 articles labeled by PMC as "research articles" were classified as a non-research type by PubMed (and thus, iCite)—in these cases, we maintained the definition of research by iCite to remain internally consistent. We used 2018 journal impact factors made available by InCites Journal Citation Reports of WOS (the latest available at the time). We used the categorizations of PMC articles across fields of science provided by the galaxy of science developed by SciTech (author: K.W.B.) [39]. Briefly, this approach clusters similar articles together and allocates each cluster to the field of the dominant journal within that cluster. Note that using this approach, it may happen that, for example, articles from medical journals are labeled as "Chemistry" if they end up in a cluster dominated by articles from chemistry journals. Also note that this galaxy of science is based on PubMed, for which reason articles found on PMC but not on PubMed have not been given a field allocation—this also applies to OCC.

## Manual assessment of transparency and reproducibility across 499 PubMed articles (2015 to 2018)

**Extraction of article characteristics.** For each article, we gathered the following information: PMID, PMCID, title, authors, year of publication, journal, first author country of affiliation, field of study (as provided by WOS; more information in our protocol [40]), and type of publication (e.g., research article, review article, case series, etc.) as detailed in our protocol [40].

**Extraction of indicators of transparency and reproducibility.** Two reviewers (S.S. and D.G.C.I.) used a previously published protocol from Wallach and colleagues [40] to extract appropriate information from eligible articles.

We first extracted any mention of a COI or funding disclosures from all abstracts and full-text articles in English. Then, for all English records with empirical data (henceforth referred to as "research articles"), we further extracted whether (a) the abstract and/or full-text of each eligible record mentions any protocol registration (whether for the whole study or part of the study), data sharing, or code sharing; and whether (b) the abstract and/or introduction of each eligible record implies any novelty (e.g., "our study is the first to identify this new protein," etc.) or that at least part of this study represents a replication of previous work (e.g., "our study replicates previously reported results in a new population," etc.). We further extracted whether (a) COI disclosures mentioned any conflict or not; (b) disclosures of funding mentioned any of public or private funds; and (c) whether websites included within data sharing statements were indeed accessible. In addition to the data extracted on the basis of the aforementioned protocol, for each one of the extracted indicators, we also extracted the text in which the information was identified to facilitate our work on automated extraction of these indicators. We only considered clear statements of these indicators, did not attempt to identify whether these statements were complete (e.g., did the authors report all of their conflicts of interest?) or truthful (e.g., has this finding truly never been published before?), and did not consider statements that were not included in the PubMed site or full text.

Note that as per the Wallach and colleagues protocol and in a deviation from Iqbal and colleagues, which counted all studies with Supporting information as potentially containing a partial/complete protocol, in this study, we downloaded and examined all Supporting information to verify whether they indeed contain any of data, code, or protocol registration. Given differences in languages between fields, it should be noted that by protocol registration, we refer to active preregistration and public availability of a study protocol, such as those found for clinical trials on ClinicalTrials.gov.

Note also that we found the Novelty and Replication indicators particularly ambiguous, for which reason we have created a document with further specifications of nontrivial cases (S1 Text). After compiling this document, we proceeded to have both main reviewers (S.S. and D.G.C.I.) reassess and cross-check all of their articles, to reduce variability in labeling due to systematic reviewer differences.

Information about 40 randomly identified articles was extracted by 3 reviewers (S.S., D.G.C.I., and J.D.W.). Upon studying discrepancies and clarifying aspects of the protocol, 2 reviewers (S.S. and D.G.C.I.) extracted relevant information, each from 240 articles. Any uncertainties were discussed between all 3 reviewers to maintain a homogeneous approach. Discrepancies were identified in adjudication of novelty and replication, for which reason these indicators were re-extracted for each article, and all unclear articles were discussed between reviewers. Information about concordance is available as a Supporting information (S10 Table). All extracted data were harmonized into a unified database, which can be accessed on OSF at https://doi.org/10.17605/OSF.IO/E58WS.

## Automated assessment of transparency: Development

We adjusted a previously reported algorithm developed by N.R. [29] to identify data and code sharing and developed algorithms to identify COI disclosures, funding disclosures, and protocol registration statements. All algorithms were constructed to take a PDF file of a typical article available on PubMed and output (a) whether it has identified a statement of relevance; (b) why it is of relevance; and (c) what the exact phrase of relevance is. This flexibility was built in (a) to help integrate these algorithms into the researcher workflow by combining manual and automated inspection; (b) to allow for different definitions of the indicators by different investigators (e.g., consider only COI disclosures that were specifically included as a stand-alone statement, rather than within acknowledgments); and (c) to ease adjudication of their performance. All algorithms, including those for data and code sharing, were further adopted to work with XML files from PMC (i.e., using the NLM XML structure).

Before using any of the algorithms, we preprocessed the text to fix problems with text extraction from PDF files (e.g., inappropriately broken lines, non-UTF8 symbols, etc.), remove non-informative punctuation (e.g., commas, full stops that do not represent the end of a phrase [e.g., "no. 123"], etc.), and remove potentially misleading text (e.g., references, etc.). For COI disclosures, Funding disclosures, and Protocol registration, text was not converted to lower or uppercase, we did not use stemming (e.g., we did not convert "processing" into "process"), and we did not remove stop words (e.g., "and," "or," etc.)—even though these are frequent preprocessing steps in natural language processing, we found these nuances informative and exploitable. Text was tokenized into phrases for data and code sharing and tokenized into paragraphs for all other algorithms; for the algorithms we developed from scratch, we used a custom-made tokenizer because already available tokenizers were not found to be accurate or flexible enough. Even though we also considered using machine learning approaches to extract these indicators, we found that the current approach performed well and afforded a level of interpretability and flexibility in definitions that is not easily achievable by alternative methods.

All programming was done in R [41] and particularly depended on the packages tidyverse [42], stringr [43], and xml2 [43,44].

The programs were structured into several different kinds of functions. Helper functions ($n = 7$) were developed to help in creating complex regular expressions more easily and in dealing with XML files. Preprocessing functions ($n = 9$) were developed to correct mistakes introduced by the conversion from PDF to text and turn the text into as conducive a document to text mining as possible. Masking functions ($n = 7$) were developed to mask words or phrases that may induce mislabeling (e.g., in searching for funding disclosures, we are masking mentions of finances within COI disclosures to avoid mislabeling those statements as funding disclosures). Labeling functions ($n = 81$; 20 for COI, 39 for Funding, and 22 for Registration) used regular expressions to identify phrases of interest (described below). The regular expressions of these labeling functions also take into account transformations of the text to improve performance (e.g., labeling functions can capture both "we report conflicts of interest" and "conflicts of interest are reported"). Localization functions ($n = 3$) were developed to identify specific locations within the text (e.g., acknowledgments). Labeling functions that were more sensitive were only applied within small localized sections of the text to reduce mislabeling. Negation functions ($n = 7$) were developed to negate potentially false labels by the labeling functions. XML functions ($n = 17$) were developed to preprocess and take advantage of the NLM XML structure. A dictionary was constructed with all phrases and synonyms used by the regular expressions ($n = 637$).

**Data and code sharing.** A member of our team (N.R.) had already developed algorithms to automatically extract information about data and code sharing [29]. Briefly, these algorithms use regular expressions to identify whether an article mentions (a) a general database in which data are frequently deposited (e.g., "figshare"); (b) a field-specific database in which data are frequently deposited (e.g., dbSNP); (c) online repositories in which data/code are frequently deposited (e.g., GitHub); (d) language referring to the availability of code (e.g., "python script"); (e) language referring to commonly shared file formats (e.g., "csv"); (f) language referring to the availability of data as a supplement (e.g., "supplementary data"); and (g) language referring to the presence of a data sharing statement (e.g., "data availability statement"). It finally checks whether these were mentioned in the context of positive statements (e.g., "can be downloaded") or negative statements (e.g., "not deposited") to produce its final adjudication. This adjudication (a) indicates whether a data/code sharing statement is present; (b) which aspect of data sharing was detected (e.g., mention of a general database); and (c) extracts the phrase in which this was detected. In this study, these algorithms were customized to avoid text in tables and references and run faster—this is the version of the algorithms that was used to study the PMCOA. It should be noted that, unlike in our manual assessment, these algorithms were designed to capture data or code sharing where new data are actually made available and avoid claims of data sharing, such as "upon request."

**Conflict of interest disclosures.** Briefly, our approach recognizes COI disclosures using regular expressions to identify whether a publication mentions (a) phrases commonly associated with a COI disclosure (e.g., "conflicts of interest," "competing interests," etc.); (b) titles of sections associated with a COI disclosure (e.g., "Conflicts of Interests," "Competing Interests," etc.); (c) phrases associated with COI disclosures (e.g., "S.S. received commercial benefits from GSK," "S.S. maintains a financial relationship with GSK," etc.); (d) phrases associated with declaration of no COI (e.g., "Nothing to disclose.," "No competing interests.," etc.); and (e) acknowledgment sections containing phrases with words associated with COI disclosures (e.g., "fees," "speaker bureau," "advisory board," etc.).

**Funding disclosures.** Briefly, our approach recognizes funding disclosures using regular expressions to identify whether a publication mentions (a) phrases commonly associated with

a funding disclosure (e.g., "This study was financially supported by . . .," "We acknowledge financial support by . . .," etc.); (b) titles of sections associated with a funding disclosure (e.g., "Funding," "Financial Support," etc.); (c) phrases commonly associated with support by a foundation (e.g., "S.S. received financial support by the NIH," etc.); (d) references to authors (e.g., "This author has received no financial support for this research.," etc.); (e) thank you statements (e.g., "We thank the NIH for its financial support.," etc.); (f) mentions of awards or grants (e.g., "This work was supported by Grant no. 12345," etc.); (g) mentions of no funding (e.g., "No funding was received for this research"); and (h) acknowledgment sections containing phrases with relevant words (e.g., "funded by NIH," etc.). This algorithm was also designed to avoid mentions of funding related to COI disclosures (e.g., "S.S. has financial relationships with GSK," etc.).

**Protocol registration statements.** Briefly, we recognize registration statements using regular expressions developed to identify the following: (a) mentions of registration on ClinicalTrials.gov and other clinical trial registries (e.g., "This study was registered on ClinicalTrials.gov (NCT12345678)," etc.); (b) mentions of registration on PROSPERO (e.g., "This study was registered on PROSPERO (CRD42015023210)," etc.); (c) mentions of registration of a protocol or a study regardless of registry (e.g., "Our protocol was registered on the Chinese Clinical Trials Register (ChiCTR-IOR-12345678)," etc.); (d) mentions of research being available on a specific register regardless of registry (e.g., "Our research protocol is available on the ClinicalTrials.gov registry (NCT12345678)," etc.); (e) titles commonly associated with registration regardless of registry (e.g., "Registration Number," "Trial registration: NCT12345678," etc.); (f) previously published protocols of studies (e.g., "Our study protocol was previously published (Serghiou et al. 2018)," etc.); and (g) registration statements within funding disclosures (e.g., "Funded by the NIH. SPECS trial (NCT12345678)," etc.). This algorithm was developed to specifically avoid mentions of registry or registration that were not relevant (e.g., "This study enrolled patients in our hospital registry.," etc.) or registrations with no open protocol availability (e.g., "Our protocol was approved by the IRS (registration no. 123456)").

## Automated assessment of transparency: Validation across 6,017 PMC articles (2015 to 2019)

**Data acquisition.** The 6,017 records obtained from PMC were used to (a) calibrate the algorithms developed in the initial dataset to the PMC dataset and (b) test the algorithms in previously unseen data. We then proceeded by using an importance sampling approach. First, we tested all 6,017 records using the algorithms developed in the training set of 499 manually assessed articles. Then, for each algorithm, we manually assessed 100 of the 6,017 articles predicted positive and 100 of the articles predicted negative. Note that the articles predicted positive or negative were different for each algorithm, hence each algorithm was not necessarily assessed in the same articles. Separately for articles predicted positive or negative, if at least 1 mistake was found, we held out 225 articles of the unseen data as a test set and redeveloped the algorithm in the remaining data until no meaningful improvement was seen. A flowchart has been made available as a Supporting information (S3 Fig). In the case of registration, only 261 out of 6,017 articles were predicted positive, for which reason we only held 161 articles as a test set. More details on our approach can be found in the Supporting information (S2 Text). All of these samples and validations have been made available (see Data Sharing statement).

**Data and code sharing.** All algorithms were tested in a sample of the PMC that had not been seen before testing (the test set, algorithm predictions, and our manual assessment are all openly available; see the Data Availability Statement). Data sharing was evaluated in 100/764

research articles predicted to share data and 89/5,253 predicted to not share data (less than 100 because not all articles were research articles). Code sharing was evaluated in 117/117 articles predicted to share code, in 93/764 articles predicted to share data, and in 88/5,253 articles predicted to not share data (7 of data sharing and 1 of non-data sharing articles had been included in the code sharing dataset).

**COI and funding disclosures.** The COI and Funding disclosure algorithms were redeveloped in the articles predicted negative for each of the 2 algorithms, respectively, out of 6,017. To do so, we set aside 225 articles and then redeveloped the algorithms in the remaining data. When confident that further development would not meaningfully improve performance, we retested each respective test set of 225 articles with the respective algorithm. As such, COI disclosures were evaluated in 100/4,792 articles predicted positive and 226/1,225 articles predicted negative (1 extra was mistakenly included and not subsequently removed to avoid bias). Funding disclosures were evaluated in 100/5,022 predicted positive and 225/995 predicted negative.

**Protocol registration.** For Protocol registration, we used a modified approach because it was very scarcely reported, as only 261/6,017 articles were predicted positive. As such, the algorithm was redeveloped in the first 100 articles predicted positive and tested in the remaining 161. Second, out of 5,756 articles predicted negative, it was very unlikely that the general article would be a false negative. As such, we employed an importance sampling approach (similar to that for code sharing), such that for each article we identified whether (a) it mentions any words of relevance to protocol registration (e.g., "registration," "trial," etc.); (b) whether it mentions any words related to the title of a methods section (e.g., "Methods," "Materials and methods," etc.); (c) whether it mentions an NCT code, which is the code given to randomized controlled trials on ClinicalTrials.gov; and (d) whether the redeveloped algorithm relabeled a previously predicted negative article as positive. Out of 5,756 predicted negative, we used a stratified/importance sampling procedure by sampling 21/3,248 articles deemed irrelevant (i.e., do not mention the words regist\*/trial/NCT), 9/451 that were deemed relevant, 44/1,951 out of those that were deemed relevant and had a Methods section, 58/91 that contained an NCT identification number, and all 15/15 that the new algorithm predicted as positive. A detailed breakdown is presented in Table 3.

## Transparency across the open access biomedical literature

First, we downloaded all of the PMCOA with clear commercial and noncommercial use labels in XML format from the PMC FTP Service [36]. Then, we processed all articles in batches of 4,096 articles and in parallel across 8 CPU cores to check for the aforementioned transparency

**Table 3. Number of articles sampled out of 5,756 predicted negative for protocol registration across strata of subcategorizations.**

| Is relevant | Has methods | Has NCT | True after update | Sample (*n*) | Total (*N*) |
|---|---|---|---|---|---|
| No | - | - | No | 21 | 3,248 |
| Yes | No | No | No | 9 | 451 |
| Yes | No | No | Yes | 1 | 1 |
| Yes | No | Yes | No | 9 | 24 |
| Yes | No | Yes | Yes | 1 | 1 |
| Yes | Yes | No | No | 44 | 1,951 |
| Yes | Yes | No | Yes | 11 | 11 |
| Yes | Yes | Yes | No | 49 | 67 |
| Yes | Yes | Yes | Yes | 2 | 2 |

indicators and extract metadata. Running the COI, Funding, and Protocol registration algorithms together in this fashion led to a mean processing time of 0.025 seconds per paper; running the Data and Code sharing algorithms together in this fashion led to a mean processing time of 0.106 seconds per paper. We finally combined the extracted data with data from OCC (citation counts and whether this is a research article) and SciTech field of science.

Note that certain variables from PMCOA were not reported in a consistent standardized form (e.g., country could be reported as "USA," "United States," "US," "Texas, USA," etc.)— we corrected all variations that occurred 20 or more times into a standardized form (e.g., if "Texas, USA" occurred 20 or more times, we changed it into "USA")—this is a time-consuming process, and we do not believe that standardizing variations that occur less than 20 times would meaningfully change the results presented. Data were standardized to the most common label to start with (i.e., if "BMJ" occurred more commonly than "British Medical Journal," then the two were standardized into "BMJ"), apart from countries, which were standardized to the name given in ggplot2 [45], which is an R package that we used to create maps. All maps are in the public domain (http://www.naturalearthdata.com/about/terms-of-use/).

We used these data to create univariable descriptive statistics and frequency counts across journals, publishers, country of affiliation, and field of science, as they were reported by PMCOA and SciTech. Unlike the preplanned analysis of indicator prevalence and distribution across time and field, all other analyses were exploratory. To mitigate data dredging inherent to such exploratory analyses, these analyses were developed in a random sample of 10,000 records and then applied to all data.

All variables considered were presence and text of COI/funding disclosures, registration statements, title, authors, affiliations, journal, publisher, type of article, references, number of figures, number of tables, citations, and field of science. Our a priori associations of interest were the distribution of indicators across time and across fields of science, as they are defined by SciTech.

It should be noted that certain journals (e.g., *Scientific Reports* and *Nature Communications*) make conflicts of interest disclosures available in the published PDF version of an article, but do not always include those in the XML versions. As such, our estimate of the presence of conflicts of interest in the open biomedical literature is a valid estimate of which statements are included in the PMC version of the text, but a relative underestimation in terms of what statements are included in the print versions.

## Statistical information

Homogeneity between the 2 reviewers as well as with the previous reviewer (J.W.) was assessed by quantifying the frequency of identification of each feature by each reviewer. These frequencies were statistically compared using Fisher exact test of independence and calculating a 2-sided *p*-value—we did not use common measures of inter-rater reliability because each reviewer assessed a different random batch of articles. Validation of the automated feature extraction algorithms was evaluated using accuracy, sensitivity (= recall), specificity, PPV and NPV (PPV = precision), prevalence of the indicator, and error between estimated and true prevalence (in terms of absolute difference) (for a detailed explanation of definitions and procedures, see S4 Text). The 95% confidence interval around the diagnostic metrics was built using the nonparametric bootstrap with 5,000 iterations and taking the 2.5th and 97.5th quantiles—in building this confidence interval, we considered the variability introduced by all sampling steps (i.e., sampling 6,017 from PMC and sampling 225 from those predicted positive or negative). For the whole PMCOA, we produced univariable frequency statistics for all variables and frequency statistics of each indicator variable across years, journal, publisher, country, and

field of science. The estimate of indicator prevalence was adjusted by considering the observed PPV and NPV in the test set, such that for an observed prevalence $p$, the adjusted prevalence was $p{\times}PPV+(1-p){\times}(1-NPV)$. $p$-Values were produced using nonparametric tests (Kruskal–Wallis test for continuous data and Fisher exact test for discrete). Correlation coefficients were calculated using the Spearman correlation coefficient.

## Data sharing

All data are available on the OSF and may be accessed, shared, or adapted under the Creative Commons Attribution 4.0 International License at the following link: https://doi.org/10.17605/OSF.IO/E58WS.

## Code sharing

All code is available on GitHub at https://github.com/serghiou/transparency-indicators/, and our algorithms are available under a GNU-3 license as an R package called rtransparent on GitHub at https://github.com/serghiou/rtransparent.

## Supporting information

**S1 Fig. Types of data sharing statements and funding disclosures in 349 research articles (2015–2018).** Of 68 research articles with Data sharing statements, most claimed availability upon request or made use of public data. Of those actively sharing new data, most made their data available on an online repository (e.g., GenBank); 8 articles stated that all of their data were available in the text or supplements, but we could not locate any such raw data—all 8 were published in *PLOS ONE*; 4 articles only shared PCR primers; 4 articles actively indicated that they are not currently sharing their data. Of 284 research articles with Funding disclosures, most reported public funds (e.g., NIH) or funds from NGOs (e.g., Gates Foundation). Very few indicated no or private funding. The data underlying this figure can be found on OSF at http://www.doi.org/10.17605/OSF.IO/E58WS. NGO, Non-Governmental Organization; NIH, National Institutes of Health; OSF, Open Science Framework.
(TIF)

**S2 Fig. Example predictions.** This figure illustrates examples of text predicted Positive (i.e., containing the indicator of interest) or Negative (i.e., not containing the indicator of interest) by the algorithm developed in the initial sample of 499 articles and the updated algorithm using data from the 6,017 articles. Both correct, this text was labeled correctly by both algorithms; Updated correct, this text was labeled correctly only by the updated algorithm; Both wrong, this text that was labeled incorrectly by both algorithms. Notice that the green statements for COI disclosures, Funding disclosures, and Protocol registration are very explicit about their content, the orange statements slightly less so, and the red statements even less explicit—this illustrates how the algorithms were updated to capture more of the less explicit statements (see S2 Text). Note that these statements were purposefully selected because they are small and clearly exemplify the points made—to access the complete evaluation of these algorithms and all sentences classified correctly or incorrectly, please see our data on OSF at http://www.doi.org/10.17605/OSF.IO/E58WS. COI, Conflict of interest; OSF, Open Science Framework.
(TIF)

**S3 Fig. Algorithm validation flowchart.** A flowchart illustrating the basic outline of our approach to validating our algorithms.
(TIF)

**S1 Table. Indicator identification using the full-text vs. the PubMed record of an article from PubMed published between 2015 and 2018.** Novelty or Replication statement, language suggesting that the authors are claiming novelty and/or replication (Yes), or neither (No). COI, Conflict of interest; Full-text, full-text article; PubMed, the PubMed record of an article. (DOCX)

**S2 Table. Indicators of transparency across 3 different random PubMed samples studying articles from 2000 to 2014, 2015 to 2017, and 2015 to 2018 (current publication).** COI, Conflict of interest; *N*, number of all articles; *n*, number of research articles; NGO, Non-Governmental Organization. (DOCX)

**S3 Table. Indicator prevalence by presence or absence from PMC.** *p*-Values were calculated using the Fisher exact test. Non-PMC, articles available on PubMed, but not PMC; PMC, articles available on PubMed and PMC; *N*, all articles; *n*, research articles. (DOCX)

**S4 Table Descriptive statistics for all metadata variables.** (HTML)

**S5 Table Descriptive statistics for all indicators of transparency-related variables.** (HTML)

**S6 Table. The 10 most common combinations of indicator co-occurrence in all 2,751,420 open access PubMed Central (PMCOA) publications.** PMCOA, PubMed Central Open Access. (DOCX)

**S7 Table. Metadata across indicators from 2,498,496 articles published from 2000 onwards.** The *p*-value was calculated using the Kruskal–Wallis nonparametric test (tests whether samples originate from the same distribution)—for our data, this is equivalent to the Mann–Whitney U test. *p*-Values <10–16 have been replaced by <10–16. All values are per article. Citation count was taken from OCC and Journal Impact Factor of 2018 from WOS. OCC, Open Citation Collection; WOS, Web of Science. (DOCX)

**S8 Table. Representative clusters, journals, and reviews for each field of science.** This table was created like so: First, we only kept clusters with at least 250 articles between 2015 and 2019. Then, we randomly sampled 3 clusters within each field. We then identified the name of each cluster, the most prevalent journal within that cluster, and the title of a representative review for each cluster. Each cluster, journal, and name within each field has been separated by a semi-colon. Two fields were excluded (EARTH and HUMANITIES) due to the 250 papers threshold—these fields have very little presence in PubMed. (DOCX)

**S9 Table. Representative text for each indicator of transparency across fields of science.** These phrases were chosen like so: First, a random sample of 10 articles was identified for each field-indicator pair (seed 1515). Then, the sample was ordered in terms of descending citation counts and was inspected from top to bottom for succinct representative phrases. The text was truncated to fit in 3 lines and any truncated text is denoted as "[...]". The extracted text for "Code sharing" in Humanities is a false positive—2 (0.5%) texts from the Humanities were labeled as sharing code, both of which were false positives; nevertheless, both texts were highly

relevant to code. For the code and data used to construct this table, see Data and Code Sharing statements.
(DOCX)

**S10 Table. Reviewer concordance.** Reviewer concordance was very good between the 2 new reviewers (S.S. and D.G.C.I), as well as the 2 reviewers and the previous reviewer (J.D.W.). Appreciable deviations were only seen in the assessment of Novelty, presence of a Replication component, and Funding disclosures, the latter of which reached statistical significance (95% CI, 0%–18%). All 3 of these were manually re-extracted and adjudicated by both reviewers to ascertain that these discrepancies did not reflect systematic differences in extraction.
(DOCX)

**S1 Text. Specifications of nontrivial cases for Novelty and Replication indicators.**
(DOCX)

**S2 Text. Code sharing comments.**
(DOCX)

**S3 Text. Associations between indicators and literature characteristics.**
(DOCX)

**S4 Text. Explanation of definitions and procedures for the validation of automated feature extraction algorithms.**
(DOCX)

## Acknowledgments

We would like to thank the National Library of Medicine, Altmetric, Crossref and the open source community for creating the open data and tools that made this work possible.

## Author Contributions

**Conceptualization:** Stylianos Serghiou, Despina G. Contopoulos-Ioannidis, Kevin W. Boyack, Joshua D. Wallach, John P. A. Ioannidis.

**Data curation:** Stylianos Serghiou, Despina G. Contopoulos-Ioannidis, Kevin W. Boyack.

**Formal analysis:** Stylianos Serghiou, Kevin W. Boyack.

**Funding acquisition:** Stylianos Serghiou, Kevin W. Boyack, John P. A. Ioannidis.

**Investigation:** Stylianos Serghiou, Despina G. Contopoulos-Ioannidis, Kevin W. Boyack, Joshua D. Wallach.

**Methodology:** Stylianos Serghiou, Despina G. Contopoulos-Ioannidis, Joshua D. Wallach, John P. A. Ioannidis.

**Project administration:** Stylianos Serghiou, John P. A. Ioannidis.

**Resources:** Stylianos Serghiou, Kevin W. Boyack.

**Software:** Stylianos Serghiou, Nico Riedel.

**Supervision:** John P. A. Ioannidis.

**Validation:** Stylianos Serghiou, Despina G. Contopoulos-Ioannidis, Kevin W. Boyack, Joshua D. Wallach.

**Visualization:** Stylianos Serghiou, Kevin W. Boyack.

**Writing – original draft:** Stylianos Serghiou.

**Writing – review & editing:** Stylianos Serghiou, Despina G. Contopoulos-Ioannidis, Kevin W. Boyack, Nico Riedel, Joshua D. Wallach, John P. A. Ioannidis.

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
