## [Editor Report · Decision Letter 0]

10 Nov 2020

Dear John, 

Thank you for submitting your manuscript entitled "Αssessment of transparency indicators across the biomedical literature: how open is open?" for consideration as a Meta-Research Article by PLOS Biology.

Your manuscript has now been evaluated by the PLOS Biology editorial staff, as well as by an academic editor with relevant expertise, and I'm writing to let you know that we would like to send your submission out for external peer review.

Please re-submit your manuscript within two working days, i.e. by Nov 12 2020 11:59PM.

Best wishes,

Roli

Senior Editor

PLOS Biology

---

## [Decision Letter · Decision Letter 1]

11 Dec 2020

Dear John,

Thank you very much for submitting your manuscript "Αssessment of transparency indicators across the biomedical literature: how open is open?" for consideration as a Research Article by PLOS Biology. As with all papers reviewed by the journal, yours was evaluated by the PLOS Biology editors as well as by an Academic Editor with relevant expertise and by four independent reviewers.

You'll see that all four reviewers were broadly positive about your study, and the concerns they raise are largely textual and/or presentational (though we do draw your attention to reviewer #1's difficulty with the R package, which must be rectified). Based on the reviews, we will probably accept this manuscript for publication, assuming that you will modify the manuscript to address all of the points raised by the reviewers. IMPORTANT: Please also make sure to address my Data Policy-related requests noted at the end of this email.

We expect to receive your revised manuscript within two weeks. Your revisions should address the specific points made by each reviewer. 

-  a cover letter that should detail your responses to any editorial requests, if applicable

*Published Peer Review History*

*Early Version*

Best wishes,

Roli

Senior Editor,

rroberts@plos.org,

PLOS Biology

DATA POLICY:

Many thanks for depositing the raw data and code in OSF. However, we also ask that all individual quantitative observations that underlie the data summarized in the figures and results of your paper be made available in one of the following forms:

Regardless of the method selected, please ensure that you provide the individual numerical values that underlie the summary data displayed in the following figure panels as they are essential for readers to assess your analysis and to reproduce it: Figs 1AB, 2, 3AB, 4, 5AB, 6, S1. NOTE: the numerical data provided should include all replicates AND the way in which the plotted mean and errors were derived (it should not present only the mean/average values).

INPORTANT: Please also ensure that figure legends in your manuscript include information on where the underlying data can be found, and ensure your supplemental data file/s has a legend.

REVIEWERS' COMMENTS:

Reviewer #1:

[identifies himself as Adrian Barnett]

This was an interesting paper describing a worthwhile task of automating assessing research papers for key transparency criteria. Transparency is a vital tool for improving the currently low quality of much medical research. Having these statistics easily available will make it possible to identify researchers/journals/institutions that are doing relatively well or badly.

The authors have put a lot of hard work into developing the algorithms to classify papers, including robust validation by hand. It is an impressive achievement to get such high sensitivity/specificity figures. I think these tools will be useful for research and will likely be improved as people use them.

Sadly, I could not get the R package to work on some test papers that I downloaded. A vignette and/or more instructions would be useful. My faith in the tools would be greatly increased if I could see them working on my own system.

The results showed surprisingly high figures for transparency, particularly on data sharing. There were also encouraging increasing trends over time for conflicts of interest. The other indicators were relatively flat, demonstrating - yet again - how science needs to improve.

The data sharing indicator includes researchers promising to make their data available, but researchers often fail to follow-through on these promises and the best level of data sharing is a link to the actual data. However, at least this is able to document the presence of a promise and there may be later consequences, such as when a researcher claims no conflict of interest but then one is found. 

Minor comments

- You could simply round percents over 10% to whole digits, except the sensitivity/specificity percents near 100. See "Too many digits: the presentation of numerical data", DOI: 10.1136/archdischild-2014-307149

- Some false positives could well occur for articles that are opinion pieces about data or code sharing.

- An agreement statistic would be more appropiate than Fisher's test for comparing raters. E.g., Gwet's agreement statistic.

- You mention standardisation at the NLM, it would also be useful to encourage journals to use standard wording around data sharing, conflicts, etc. Although that is a much harder task.

Reviewer #2:

[identifies herself as Quinn Grundy, University of Toronto]

The authors aimed to develop an algorithm capable of assessing multiple indicators of transparency (conflict of interest and funding disclosures) and reproducibility (data sharing, code sharing and protocol registration statements) across the biomedical literature indexed in PubMed at scale. Their algorithm largely demonstrated accuracy, sensitivity and specificity across indicators, with the exception of indicators related to data sharing and code sharing. Overtime, they demonstrate that the proportion of articles including an indicator has increased for all indicators over the study period and particularly, conflict of interest and funding disclosures. Overall, the paper was well-written and comprehensive in its reporting. 

I make the following comments, but please note that my expertise relates to conflicts of interest and research integrity and I am not qualified to evaluate the methods related to the building or testing of the algorithm.

The authors state that the chief contribution of the paper is the algorithm and corresponding data set, which they make open access. However, I think the contribution of this work and the algorithm/data set could be more explicitly stated both in the introduction and in the discussion.

In the introduction, the authors make a case for achieving transparency and more open science. Transparency is often the reflexive reaction and 'fix' suggested for all issues related to research integrity, however, it really is a means to an end. The introduction could be strengthened by distinguishing and making more explicit the relationships among the key concepts (transparency, reproducibility and replication) and their relationship to research integrity. 

In the discussion, the authors state that their work demonstrates that "using automated approaches to study and understand the biomedical literature is possible and can yield insights over and above those possible with manual assessments of small random samples." However, they also note that uptake of simple interventions such as XML tagging has been highly limited. The authors might consider more explicitly denoting the audience for this work (researchers?) or suggesting particular implementations (or noting that this work remain exploratory). For example, could this dataset be used by journals or fields of science to benchmark their transparency practices? In what ways could these tools be used to understand actual transparency and reproducibility rather than indicators?

The findings of this study also need to be contextualized beyond the 'biomedical' literature. The authors included all articles indexed in PubMed; from the Tables of study characteristics, it is apparent that articles from a number of fields of study including humanities, 'health', and social sciences were included. For a number of these fields of study, empirical articles may include studies for which some of the indicators are inappropriate, namely data sharing, code sharing and protocol registration (e.g. qualitative research, data that cannot be anonymized, analysis that does not involve code, designs for which you can't register protocols nor are reproducible). The authors might comment on when/how the algorithm is most applicable, where it has limited value, and where it can be universally applied (e.g. funding disclosures). Further, how did the authors determine what was considered "empirical" research? Figure 5 in particular should be contextualized.

Minor comments

For manual assessment, you report, "and a substantial portion mentioned use of public funds". Do you mean that this portion disclosed support from public funding sources only? (i.e. publicly-funded research?) Could you rephrase to clarify? 

Figure 1 - how were these data combined with the data from the previous studies? Also, I did not understand the rationale for combining these data sets. 

It was my understanding that PubMed/NLM instituted a change that required the inclusion of conflict of interest and/or funding statements in the meta-data. Can you comment on this policy change? Does this refer to the fact that many journals are not making use of the XML tags? Could this account for the notable increase in prevalence of COI statements? 

In Figure S2. I did not understand what was meant by "both" correct, "both" wrong? Could you explain?

Open Citation Collection - how do they define "research articles"? 

The authors state, "Note that unlike in our manual assessment, the numbers reported in this section refer to the entire literature on PMCOA, not merely research articles; all analyses were repeated in research articles alone with no meaningful changes." It would seem that for the vast majority of records (e.g. editorials, reviews, commentaries, letters etc.) that Data sharing and Code sharing would be irrelevant. Could the authors comment on this finding?

With funding, could discrepancies be because work was actually unfunded (and thus no disclosure)? With funding, absence of disclosure may indicate no funding. Or, conflation of COI and funding.

When you state, "Similarly, no publisher reports Data sharing in more than ~75% of publications. Is this among research articles? Or all publications (i.e. journals)? 

Can you comment as to why XML tags are so infrequently used? Who is responsible for appending these to meta-data? What infrastructure/policy would this require?

Reviewer #3:

This manuscript does an excellent job of successfully demonstrating an automated approach for measuring and characterizing the transparency of the biomedical literature. In addition to identifying important trends by time, discipline, and indicator type, the authors present a set of open algorithms and data that are intentionally designed/presented to facilitate future and more nuanced research questions.

Only a few minor revisions are recommended:

1. The authors describe alternative automated approaches for assessing transparency, but do not reference SciScore. While SciScore focuses on indicators of rigor, I believe its shared attention to the problem of reproducibility is potentially noteworthy.

2. While the authors identify the five indicators of transparency their work addresses, they do not offer a rationale for why these indicators were chosen. It would be helpful for readers who less familiar with this field to provide a brief, but explicit explanation.

3. Results section, P1, S1 - consider replacing alternative with additional.

4. In the manual review of PubMed articles, "data upon request" like statements were, I believe, considered positively indicative of transparency; however, the automated approach applies a stricter criterion. Consider briefly describing the reason for this change.

5. It would be helpful to reference the criterion OCC uses to identify "research" articles: https://icite.od.nih.gov/user_guide?page_id=ug_data#article.

6. The authors' description of their automated method for identifying pre-registered protocols strikes me as focused on repositories and naming conventions specific to clinical trials and human-subjects research; however, a significant portion of the analyzed papers are pre-clinical. If my interpretation of the methodology is accurate, consider addressing the focus and limitations of this approach. If my interpretation is inaccurate, revise the methods to provide non-clinical examples.

Reviewer #4:

[identifies himself as Fabian Held]

Thank you for giving me the opportunity to review this manuscript titled "Assessment of transparency indicators across the biomedical literature: how open is open?". It presents an open-source, automated approach to identify five indicators of transparency in biomedical research literature, including conflicts of interest disclosures, funding disclosures, data sharing, code sharing and protocol registration. The manuscript describes the development and evaluation of algorithms that automatically identify relevant parts of a publication. Furthermore, it demonstrates that this can be achieved at the scale of millions of articles by analysing the entire open access biomedical literature of 2.75 million articles on PubMed Central. 

The insights and tools presented here constitute a substantial advancement in our ability to monitor transparency in research and encourage reproducible research practices and the assessment of risk of bias.

* The work presented is complex: five different algorithms were developed, evaluated, and employed to report findings across millions of articles. Unfortunately, development for CoI and funding extraction deviated from the train-validate-test paradigm. The manuscript uses several subsets of data with various qualifications across multiple stages and the current explanation of their evaluation process is hard to follow. In fact I am unclear how the final algorithms were evaluated, how their test sets were coded and how (or if) the test sets are related across algorithms. One example is the following statement from p.31: "To do so, we set aside 225 articles and then redeveloped the algorithms in the remaining data. […] COI disclosures were evaluated in 100/4792 predicted positive and 226/1225 predicted negative (one extra was mistakenly included and not subsequently removed to avoid bias). Funding disclosures were evaluated in 100/5022 predicted positive and 225/995 predicted negative." The evaluation process in the methods section should be rewritten for clarity. Possibly a schematic diagram of the process coding and data (sub)sets could be helpful. 

* Furthermore it would be instructive for readers to mention earlier in the paper what kinds of algorithms are discussed here. These are not machine learning or statistical algorithms, but largely rules-based, using regular expressions. 

* In the results sections I would recommend to add a column for totals to Fig5a and I am unconvinced that Fig5b is an effective visualisation. Sizes of very small circles are difficult to interpret and the added colour coding for fields of science distorts the visual effect of size. Maybe shading could be used for estimated prevalence and a coloured overlay could indicate field?

* Finally, the labelling in the third table in S4 Text. Performance metrics speaks of "manual assessment" which strikes me as misleading as this is a table of expected values.

---

## [Decision Letter · Decision Letter 2]

14 Jan 2021

Dear John,

Many thanks for submitting your revised Research Article entitled "Αssessment of transparency indicators across the biomedical literature: how open is open?" for publication in PLOS Biology. I've now obtained advice from one of the original reviewers and we're nearly there; I just need you to tidy up a few final issues...

IMPORTANT:

a) Please could you address the remaining points raised by reviewer #1? Some pertain to the Github deposition, and some to the manuscript itself.

b) Many thanks for making the data underlying the Figures available in your OSF deposition. However, please could you cite the URL in all relevant main and supplementary Figure legends, e.g. "The data underlying this Figure may be found at http://www.doi.org/10.17605/OSF.IO/E58WS"

We expect to receive your revised manuscript within two weeks. Your revisions should address the specific points made by each reviewer. 

-  a cover letter that should detail your responses to any editorial requests, if applicable

*Published Peer Review History*

*Early Version*

Best wishes,

Roli

Senior Editor,

rroberts@plos.org,

PLOS Biology

REVIEWER'S COMMENTS:

Reviewer #1:

[identifies himself as Adrian Barnett]

This is an important and worthwhile piece of research and the authors have answered my previous comments. 

The authors have added a vignette and I was able to run their code on some papers and see it working. I have some minor quibbles with the vignette, but these are not a bar to publishing the paper in its current form.

There was no rendered version of the vignette online, so whilst I could run the vignette I could not check my results against the authors. 

I was able to get the PDF version working, but not the version using XML. E.g., for the example used in the vignette I got the following when using the PDF route:

$ article <chr> "10.17605/OSF.IO/E58WS"

$ is_open_data <lgl> TRUE

$ open_data_category <chr> "field-specific repository, general-purpose repository, data availability statement"

$ is_open_code <lgl> FALSE

$ open_data_statements <chr> "\"all data are made available on a public repository (openfmri accession number ds000202).; \"raw data derived from this anal...

$ open_code_statements <chr> ""

And the following when using the XML route:

$ pmid <chr> ""

$ pmcid_pmc <chr> ""

$ pmcid_uid <chr> ""

$ doi <chr> ""

$ filename <chr> "PMID30457984.xml"

$ is_research <lgl> FALSE

$ is_review <lgl> FALSE

$ is_success <lgl> TRUE

Minor comments

* Table 2, not really a "distribution" as it's simply the number and percent. A couple of large numbers are missing commas, e.g., Medicine row. 

* "Funding disclosure information in 67.9% (67.6-68.3%)," missing "95% CI"

* It's hard to see small countries in Figure 4. A table of the top 10 countries might be more useful.

* It was useful to have examples of where the code worked and did not work (Fig s2).

* The help files for rtransparent need some more detail to tell users what the variables are. For example, the command to search for funding returns 52 variables, but the help file does not detail what these are.

---

## [Editor Report · Decision Letter 3]

19 Jan 2021

Dear John,

On behalf of my colleagues and the Academic Editor, Lisa Bero, I'm pleased to say that we can in principle offer to publish your Research Article "Αssessment of transparency indicators across the biomedical literature: how open is open?" in PLOS Biology, provided you address any remaining formatting and reporting issues. These will be detailed in an email that will follow this letter and that you will usually receive within 2-3 business days, during which time no action is required from you. Please note that we will not be able to formally accept your manuscript and schedule it for publication until you have made the required changes.

PRESS: We frequently collaborate with press offices. If your institution or institutions have a press office, please notify them about your upcoming paper at this point, to enable them to help maximise its impact. If the press office is planning to promote your findings, we would be grateful if they could coordinate with biologypress@plos.org. If you have not yet opted out of the early version process, we ask that you notify us immediately of any press plans so that we may do so on your behalf.

Thank you again for supporting Open Access publishing. We look forward to publishing your paper in PLOS Biology. 

Best wishes,

Roli

Roland G Roberts, PhD 

Senior Editor 

PLOS Biology